# Ruxolitinib with resminostat exert synergistic antitumor effects in Cutaneous T-cell Lymphoma

Fani Karagianni[1]*, Christina Piperi[2], Vassiliki Mpakou[3], Aris Spathis[4], Periklis G. Foukas[4], Maria Dalamaga[1,2], Vasiliki Pappa[3], Evangelia Papadavid[1]

1 2nd Department of Dermatology and Venereal Diseases, NKUA, Athens, Greece, 2 Department of Biological Chemistry, NKUA, Athens, Greece, 3 Second Department of Internal Medicine and Research Institute, Attikon University General Hospital, NKUA, Athens, Greece, 4 Second Department of Pathology, National and Kapodistrian University of Athens, Athens, Greece

☉ These authors contributed equally to this work.
* karagiannifani@gmail.com

## Abstract

### Background

The combination of JAK/STAT and HDAC inhibitors exerted beneficial effects in haematological malignancies, presenting promising therapeutic CTCL targets. We aim to investigate the efficacy of JAK1/2i ruxolitinib in combination with HDACi resminostat in CTCL *in vitro*.

### Material & methods

Non-toxic concentrations of ruxolitinib and/or resminostat were administered to MyLa (MF) and SeAx (SS) cells for 24h. Cytotoxicity, cell proliferation and apoptosis were estimated through MTT, BrdU/7AAD and Annexin V/PI assay. Multi-pathway analysis was performed to investigate the effect of JAK1/2i and/or HDACi on JAK/STAT, Akt/mTOR and MAPK signalling pathways.

### Results

Both drugs and their combination were cytotoxic in MyLa ($p < 0.05$) and in SeAx cell line ($p < 0.001$), inhibited proliferation of MyLa ($p < 0.001$) and SeAx ($p < 0.001$) at 24h, compared to untreated cells. Moreover, combined drug treatment induced apoptosis after 24h ($p < 0.001$) in MyLa, and SeAx ($p < 0.001$). The combination of drugs had a strong synergistic effect with a CI<1. Importantly, the drugs' combination inhibited phosphorylation of STAT3 ($p < 0.001$), Akt ($p < 0.05$), ERK1/2 ($p < 0.001$) and JNK ($p < 0.001$) in MyLa, while it reduced activation of Akt ($p < 0.05$) and JNK ($p < 0.001$) in SeAx.

### Conclusion

The JAKi/HDACi combination exhibited substantial anti-tumor effects in CTCL cell lines, and may represent a promising novel therapeutic modality for CTCL patients.

**Data Availability Statement:** All relevant data are within the manuscript.

**Funding:** The author(s) received no specific funding for this work.

**Competing interests:** The authors have declared that no competing interests exist.

## 1. Introduction

Cutaneous T-cell lymphomas (CTCLs) are rare skin malignancies, forming a heterogeneous group of non-Hodgkin lymphomas derived from skin-homing mature T-cells [1]. Mycosis Fungoides (MF) is considered as the commonest type of CTCL, characterized by patches and infiltrated plaques on the skin, which eventually evolve into tumors [2, 3], whereas, Sezary syndrome (SS), the leukemic variant of MF, characterized by erythroderma, lymphadenopathy and the presence of a malignant T-cell clone in the peripheral blood and the skin [3, 4]. In advanced stages, the survival rate is poor (a median survival < 4 years, and only 13 months for those with nodal involvement) and patients often die due to an incompetent immune system [5–8].

MF and SS are considered as different diseases since they arise from distinct T-cell subsets: MF, from skin-resident effector memory T-cells (TRm) and SS, from the central memory T-cells (TCm) [9, 10]. The fundamental difference in the putative cell of origin between SS (TCM-derived) and MF (TRM-derived) is consistent with their distinct clinical behaviors, since TCm may be found in all three compartments, i.e. in peripheral blood, lymph node and skin, while resident TRm-cells remain localized at the skin. Detection of these malignant T-cell clones is critical for the diagnosis of CTCL, with TCm expressing CCR7$^+$/L-selectin$^+$ and TRm expressing CCR7$^+$/L-selectin$^-$ [11]. Recently, the genomic profiling of CTCL has been determined using high-throughput technologies, Next generation sequencing (NGS) and whole genome sequencing (WGS), to identify genetic alterations or abnormal gene expression between healthy controls and patients suffering from CTCL [12–14]. Therefore, many studies sought to explore the genetic landscape of MF/SS and have identified several mutated genes affecting cell cycle regulation, chromatin remodeling and major signal transduction pathways, including the Janus kinase (JAK)-signal transducer and activator of transcription protein (STAT), the T-cell receptor (TCR)/nuclear factor kappa-light-chain-enhancer of activated B cells (NF-κB), and the mitogen-activated protein kinase (MAPK) [14–16].

Dysregulation of the JAK/STAT pathway is implicated in the pathogenesis of solid tumors [17], as well as in some hematologic malignancies where abnormal activation of JAK2 signaling has been reported [18]. Of interest, activation of the JAK-STAT pathway has also been implicated in CTCL progression, leading to constitutive expression of STAT3 and STAT5, upregulation of cell survival (Bcl-2 and Bcl-xL) and cell cycle genes (Cyclin D, c-Myc), Th2 cytokines (IL-4) and mir-155 [19].

Several drugs inhibiting the aforementioned pathway have been shown to exhibit a therapeutic potential, including ruxolitinib (initially FDA-approved as Jakinib$^®$), a potent JAK1/2 approved for the treatment of primary myelofibrosis, and also studied in psoriasis and rheumatoid arthritis [10]. Ruxolitinib has been shown to cause a dose-dependent inhibition of cell proliferation, with concurrent activation of apoptosis, a marked and rapid inhibition of STAT activation and inhibition in DNA synthesis in CTCL cells [20].

Another class of drugs with potential modulatory effects in MF/SS pathogenesis is the family of Histone Deacetylase (HDACs) inhibitors. HDACs are implicated in a wide range of cellular processes [21–23] including JAK2/STAT3 signaling pathway inhibition through regulation of suppressor of cytokine signaling proteins (SOCS), a family of genes involved in JAK2-STAT3 signaling pathway inhibition [24, 25], while they are also highly upregulated in CTCL [26] and in B- and T- cell lymphoma patients [27].

HDAC inhibitors have been shown to interfere with cell proliferation, apoptosis and migration of neoplastic cells via the down-regulation of several pathways implicated in the activation of cytokines, chemokines, growth factors and protein kinases [28]. Two HDAC inhibitors (HDACi), vorinostat$^®$ and romidepsin$^®$ (FK228) have achieved good clinical efficacy in

CTCL with objective responses of 25–30%, leading to the FDA approval of both drugs for the treatment of CTCL in 2006 and 2009, respectively [29]. Currently, resminostat, an HDAC inhibitor targeting HDAC class I, IIb and IV is tested in a phase II clinical trial in Europe and Japan as a maintenance therapy in advanced CTCL (RESMAIN, NCT02953301).

Resminostat was reported to up-regulate a gene expression signature representative of the Th1 cell type, and down-regulate genes of the Th2 cell type, thus favouring the beneficial Th1 cell phenotype. Furthermore, resminostat exhibited a reduction at the mRNA and protein secretion of the Th2 and itch-mediating cytokine IL-31, suggesting that it might improve pruritus [30].

It can be concluded from the above studies that these monotherapies are partially effective in haematological malignancies.

In the present work, we investigated for the first time the antitumor activity of JAK2 ruxolitinib and HDAC inhibitor resminostat in the cytotoxicity, proliferation and apoptosis of CTCL cell lines, along with their impact in major cellular signaling pathways. The combined effect of these drugs was shown to target different upstream and downstream molecules in JAK/STAT pathway.

## 2. Material and methods

### 2.1 Cell lines and drugs

The human CTCL cell lines, MyLa and SeAx, were kindly provided by Dr Jan P. Nikolay (Klinik für Dermatologie, Venerologie und Allergologie, Universitätsmedizin Mannheim, Ruprecht-Karls-Universität Heidelberg, Mannheim, Deutschland), already tested and authenticated. Both cell lines were cultured in RPMI 1640, supplemented with 10% fetal bovine serum and 1% penicillin/streptomycin, and treated with/without JAK1/2 inhibitor (15μM ruxolitinib) and/or HDAC inhibitor (1 μM resminostat) at 37˚C in a humidified atmosphere with 5% $CO_2$, for 24h. The HDAC inhibitor was a generous gift from 4SC AG (Germany) and the JAK inhibitor was a gift from Novartis Incyte (USA). Both inhibitors were dissolved in DMSO according to the manufacturer's instruction. Therefore, vehicle controls or untreated cells were treated with 0.1% DMSO for all the experiments. T cell stimulation was achieved through PMA (50ng/ml) and ionomycin (1μg/ml) treatment prior to the multi-pathway analysis.

### 2.2 Cytotoxicity assay

The 3-(4,5-dimethylthiazol-2-yl)-2,5-dimethyltetrazolium bromide (MTT) assay (Thermo-Fisher Scientific) was used to measure the cytotoxicity in untreated and drug-treated CTCL cell lines. Briefly, $10^4$ cells were seeded in 96-well plates with the addition of the appropriate drugs. After 24h, the culture medium was removed and substituted with fresh one, along with MTT solution, for 4 hours and DMSO at 37˚C. The absorbance was measured at 570 nm wavelength in a microplate reader (Tecan Infinite F200, Austria).

### 2.3 BrdU assay

In this method, BrdU (an analog of the DNA precursor thymidine) was incorporated into newly synthesized DNA by cells entering and progressing through the S (DNA synthesis) phase of the cell cycle. The incorporated BrdU (BD Pharmingen) was stained with specific anti-BrdU fluorescent antibody. The levels of cell-associated BrdU were then measured by flow cytometry. Briefly, $10^5$ cells were seeded in 24-well plate for 24h with the presence/absence of specific drugs at the determined concentrations according to the manufacturer's instruction.

## 2.4 Apoptosis assay

FACS analysis using the Annexin V/PI staining method was used to determine the apoptotic rate in drug-treated and control cells. Briefly, $10^5$ cells were seeded and cultured with or without the addition of ruxolitinib and/or resminostat at the indicated concentrations for 24h. Cells were afterwards washed with ice-cold PBS and centrifuged at 500×g for 5 min. Cell pellet was re-suspended in ice-cold annexin binding buffer, and annexin-V-FITC antibody, along with propidium iodide (AssayDesigns, U.S.A.) were added and cells were incubated for 10 min in the dark. Samples were run on a Cytomics FC500 Flow Cytometry Analyzer (Beckman Coulter, USA) and cell viability and apoptosis were calculated using a quadrant on the Annexin-V-FITC/PI histogram. Living cells were negatively stained for both assays, early apoptotic cells were only Annexin-V-FITC positive, necrotic cells were only PI positive and late apoptotic cells were positively stained for both assays. The synergistic effect between resminostat and ruxolitinib was determined by the combination index (CI) based on the Chou-Talalay method [31], as previously described [32]. The CI value was determined by the following equation: CI = sum of apoptosis of single agent treatment/apoptosis upon combined treatment. The combination effect was defined as "synergistic", "additive" or "antagonistic" when CI was <1, 1 and >1, respectively. A combination index (CI) of > 1 indicates antagonism, a CI of 1 denotes additivity, and a CI of < 1 indicates synergism. More specifically, CI values ranging from 0.1–0.3 are considered to indicate strong synergism, 0.3–0.7 synergism, and 0.7–0.85 moderate synergism.

## 2.5 Multipathway panel analysis

Pellets from stimulated cells with PMA (50ng/ml) and ionomycin (1μg/ml) were collected at 24h and 48h post drug treatment, lysed and further subjected to multipathway analysis (Millipore, MILLIPLEX MAP Multi-Pathway Magnetic Bead) of JAK/STAT [p-STAT3 (ser427), p-STAT5(Tyr694/699)], Akt/mTOR [p-Akt (Ser473), p-p70S6K (Thr412)] and MAP kinase [p-ERK1/2 (Thr185/Tyr187), p-p38 (Thr180/Tyr182), p-JNK (Thr183/Tyr185) and p-NF-κB (Ser536 p65 subunit)] signalling pathways, after the aforementioned drug treatments for 24h and 48h using Luminex 200TM (Luminex Corporation Austin, TX, USA) technology, according to the manufacturer's instructions.

## 2.6 Statistical analysis

Statistical analyses were performed using the SPSS software (version 16.0; SPSS, Inc., Chicago, IL., USA). All experiments were performed in triplicates, in 3 independent repeats, and the results were presented as the mean ± standard deviation. One-way Anova and Bonferroni methods were used to compare the means among the different groups. A two-sided p value <0.05 was considered statistically significant.

# 3. Results

## 3.1 Resminostat and/or ruxolitinib induce cytotoxicity in CTCL cell lines

Different concentrations (0.3μM, 1μM and 3μM) of resminostat were initially tested in both CTCL cell lines used (S1 Fig), in order to determine cytotoxicity. Among the three concentrations used, only 1μM and 3μM resminostat significantly reduced the cell viability of MyLa and SeAx cell lines at 24h compared to cells treated with 0.1% DMSO (S1A Fig). Between the two concentrations of 1μM and 3μM resminostat, no significant differences were found. Therefore, the concentration of 1μM was selected for further combination experiments.

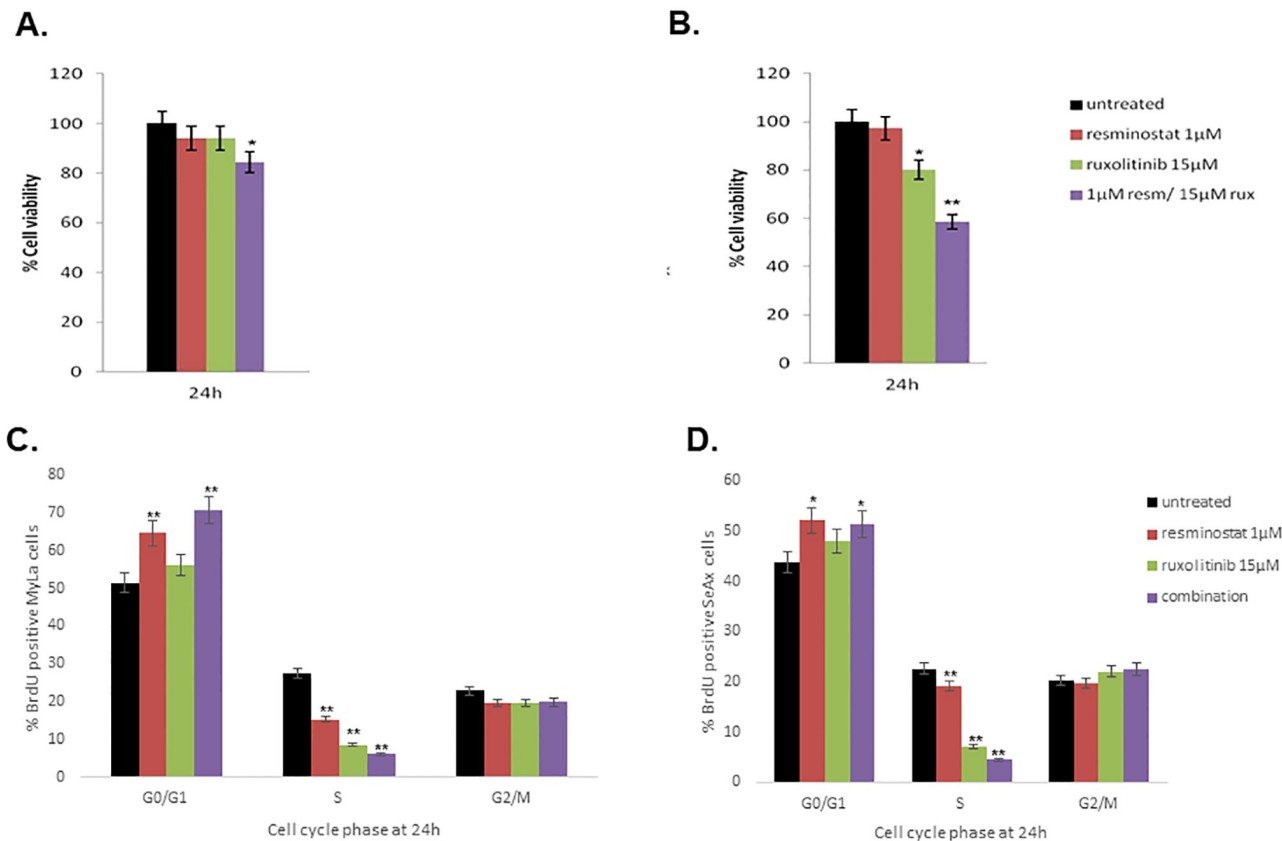

**Fig 1. Cell viability of resminostat and/or ruxolitinib in CTCL.** Effect of resminostat and/or ruxolitinib in cell cytotoxicity in Myla (A) and SeAx (B) cell lines at 24h evaluated by MTT assay. **Cell proliferation, evaluated by BrdU assay, of resminostat and/or ruxolitinib in CTCL cell lines**. G0/G1 arrest is induced by both monotherapies and combination of resminostat with ruxolitinib in MyLa (C) and SeAx (D). Values are the mean ± standard deviation of three experiments. Values are the mean ± standard deviation of three experiments. *p <0.05, **p<0.001 statistically significant differences versus untreated and single drugs or combination.

In accordance to resminostat, different concentrations (1μM, 15μM and 30μM) of ruxolitinib were also tested in CTCL cell lines (S1B Fig) Among the three concentrations used, only the concentration of 15μM and 30μM ruxolitinib significantly reduced the cell viability of MyLa and SeAx cell lines at 24h compared to cells treated with 0.1% DMSO. Between the two concentrations of 15μM and 30μM ruxolitinib, no significant differences were found. Therefore the concentration of 15μM was selected for further combination experiments.

The combination of 1μM resminostat/15 μM ruxolitinib was shown to significantly reduce the cell growth of MyLa cells (Fig 1A) after 24h (p<0.05) of treatment compared to control, whereas in SeAx cell line, a statistically significant reduction was obtained also at 24h (p<0.001) (Fig 1B).

## 3.2 Combined treatment of resminostat and ruxolitinib induces cell cycle arrest

To further investigate the inhibitory effects of resminostat and/or ruxolitinib on the proliferation of CTCL cell lines, we examined the cell cycle distribution after a 24h treatment with resminostat and/or ruxolitinib (Fig 1C and 1D). The percentages of treated MyLa and SeAx cells, with either resminostat and/or ruxolitinib was higher in G0/G1 phase compared to

control cells at significant level (p<0.001 for MyLa and p<0.05 for SeAx). There was a further increase in the percentage of the treated cells with both drugs, and this increase was even higher in the combination therapy (p<0.001). No significant changes were observed in G2/M phase. On the contrary, as expected, there was a significant decrease in S phase of both cell lines in monotherapies and combination therapy compared to untreated cells indicating that these drugs inhibit cell proliferation of MyLa and SeAx cell line at 24h of treatment.

### 3.3 Combined treatment of resminostat and ruxolitinib induces apoptosis in CTCL cell lines

Following the effect of resminostat and ruxolitinib in cell cycle proliferation, we proceeded to investigate their impact on apoptosis' induction (Fig 2).

Treatment of MyLa cells with either resminostat or ruxolitinib did not exhibit any significant effects on apoptosis after 24h (Fig 2A and 2B). However, their combination significantly increased early apoptotic (22.1%, p<0.001) and late apoptotic cell numbers (22%, p<0.001) at the 24h time point, compared to control cells (Fig 2A and 2B).

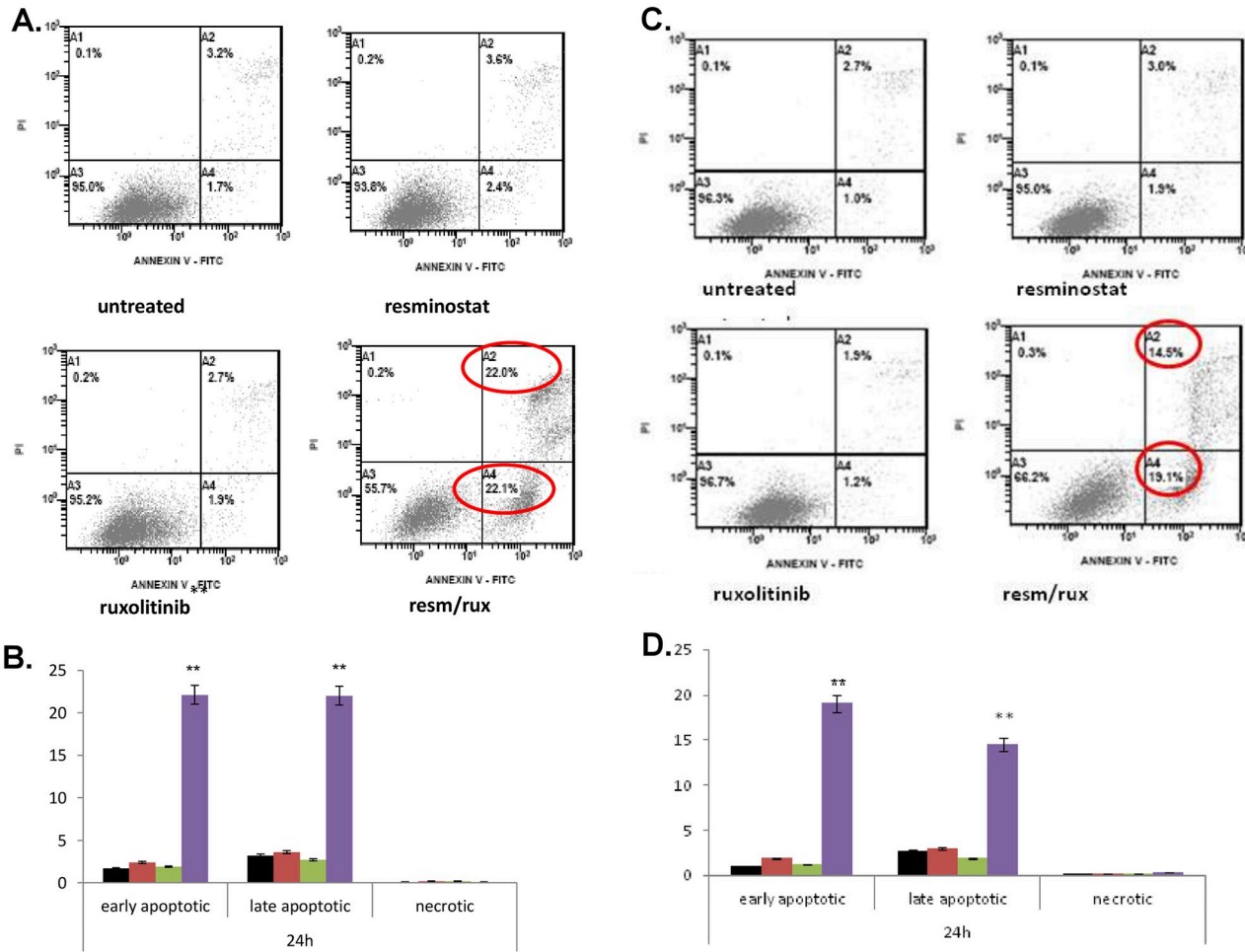

**Fig 2. Apoptotic effect of resminostat and/or ruxolitinib in CTCL cell line.** Effect of resminostat and/or ruxolitinib in cell apoptosis in Myla cell line at 24h (A, B) and SeAx cell line (C, D) after incubation with 1μM resminostat, 15μM ruxolitinib and combination of the two drugs compared to untreated. Values are the mean ± standard deviation of two experiments run in triplicates. **p<0.001 statistically significant differences versus untreated and single drugs or combination.

## Resminostat/Ruxolitinib synergistic activity

**Fig 3. Synergistic effect of resminostat and ruxolitinib in CTCL cell lines.** The CI was calculated to determine drug interactions. Effects were defined as synergistic, additive and antagonistic when CI<1, CI = 1, CI>1, respectively.

Similarly, single treatment of SeAx cells with resminostat or ruxolitinib did not seem to affect apoptotic cell numbers after 24h (Fig 2C and 2D). However, their combination significantly increased early apoptotic (19.1%, p<0.001) and late apoptotic cells (14.5%, p<0.001) at the 24h post-treatment time point, compared to control cells (Fig 2C and 2D).

The combination treatment has synergistic effects in both cell lines (Fig 3). In MyLa cells CI was 0.062 at 24h post treatment, whereas in SeAx cells, the CI was 0.141 at 24h. These values indicate strong synergism for both cell lines.

### 3.4 Inhibition of key signalling pathways after exposure to ruxolitinib, resminostat and their combination

In order to further investigate the signal transduction pathways that are activated by single or combinational resminostat and ruxolitinib treatment, we performed a multi-pathway analysis in both cell lines at two different time points.

Resminostat treatment of MyLa cells resulted in decreased Akt (p<0.001) phosphorylation levels both at 24h and 48h, compared to control cells (Fig 4B), while ruxolitinib significantly reduced p-Akt (p<0.001, Fig 4B), p-p70S6K (p<0.05, Fig 4F) and p-p38 (p<0.05, Fig 4G) levels at 24h compared to controls. Ruxolitinib treatment significantly reduced p-STAT3 (p<0.001, Fig 4A) and p-JNK (p<0.001, Fig 5D) levels at 48h post-treatment.

Interestingly, in MyLa cells the combination of the two drugs was found to significantly decrease p-STAT3 (p<0.001, Fig 4A), p-AKT (p<0.001, Fig 4B), p-ERK1/2 (p<0.045) and p-JNK (p<0.001, Fig 4D) at 24h, compared to controls. Furthermore, a significant reduction was observed at p-STAT3 (p<0.001, Fig 4A), p-AKT (p<0.001, Fig 4B), p-JNK (p<0.001, Fig 4D) and p-p38 (p<0.001, Fig 4G) at 48h post-treatment, compared to control cells.

In the SeAx cell line, resminostat treatment did not show any significant effects in the activated signaling pathways compared to control cells neither at 24h nor 48h after treatment.

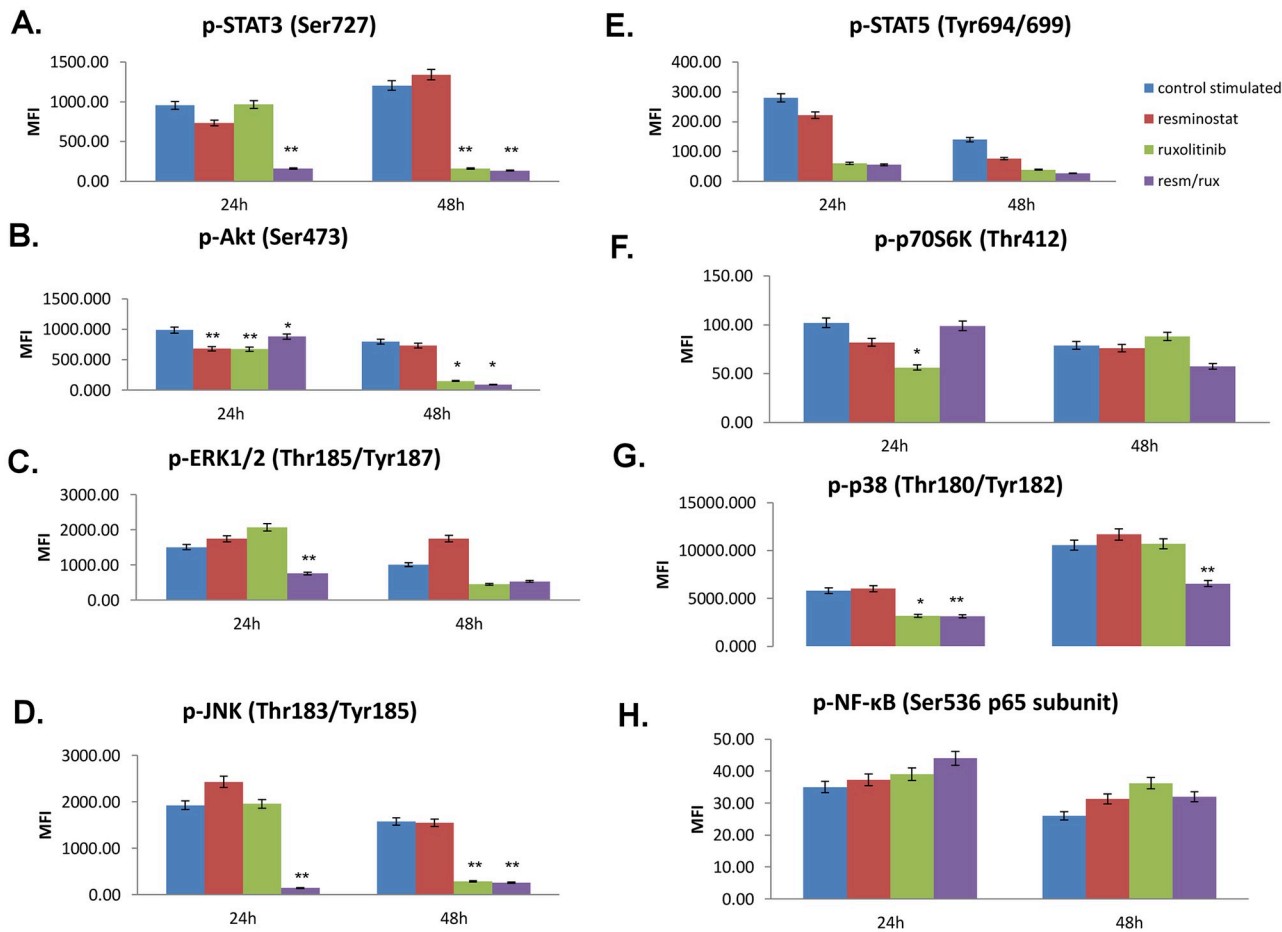

**Fig 4. Activation of key pathways in a Mycosis Fungoides cell line.** Effect of resminostat and/or ruxolitinib in the activation of key signalling pathways in Myla cells. Values are the mean ± standard deviation of two experiments run in triplicates. * p<0.05, **p<0.001 statistically significant differences versus untreated and single drugs or combination.

However, ruxolitinib treatment was shown to reduce p-JNK levels (p<0.001, Fig 5D) at 24h compared to control cells. Of note, the combination of the two drugs was found to significantly decrease p-AKT (p<0.05, Fig 5B) at 24h and at 48h and p-JNK (p<0.001, Fig 5D) only at 24h compared to controls (Fig 5).

## 4. Discussion

Current treatment approaches for MF/SS are not curative and present high rates of relapse. Patients with relapsed or refractory MF/SS display a poor prognosis, and a need for more efficient treatment options. The only curative option in MF/SS is stem cell transplantation [33]. Complete responses with existing monotherapies are still rare and data on survival under approved therapies are yet to be collected. The development of standardized approaches for the treatment of advanced disease, resistant and recurrent disease and especially standardized strategies for maintenance therapy of these patients in remission are at present a particularly interesting area of research [32]. Therefore, there is a need to develop new drugs or use of repurposed drugs that can provide more durable responses, and improve the poor outcomes of patients with advanced stage MF and SS.

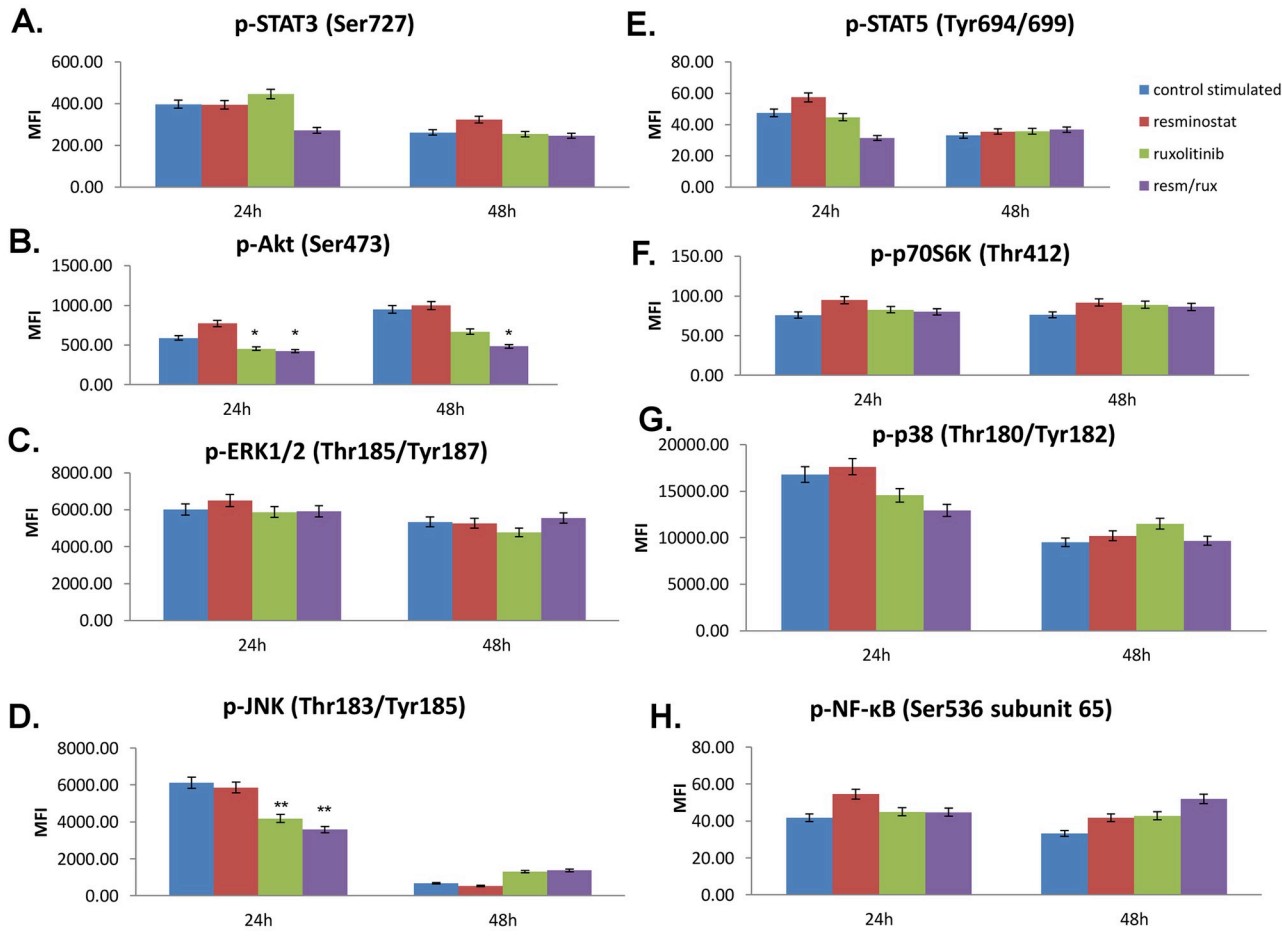

**Fig 5. Activation of key pathways in a Mycosis Fungoides cell line.** Effect of resminostat and/or ruxolitinib in the activation of key signalling pathways in SeAx cells. Values are the mean ± standard deviation of two experiments run in triplicates. * p<0.05, **p<0.001 statistically significant differences versus untreated and single drugs or combination.

Our results demonstrate for the first time that the combination of ruxolitinib, a JAK1/2i with resminostat, a HDACi demonstrates efficacy in an *in vitro* model of CTCL, highlighting its significance as a promising novel therapeutic modality for CTCL patients that fail to benefit from monotherapy.

More specifically, we found that ruxolitinib and resminostat, as monotherapies, manage to inhibit cell viability of both CTCL cell lines tested. Moreover, their combination significantly reduced the cell viability of MyLa and SeAx cells after both 24h of treatment. In line with this observation, previous studies have already shown that resminostat treatment inhibits proliferation of multiple myeloma cells [34]. Moreover, Civallero *et al*. have demonstrated the antitumor activity of ruxolitinib and vorinostat®, another HDAC inhibitor, in cell lines of hematologic malignancies, including Hodgkin's lymphoma and selected subtypes of non-Hodgkin's lymphoma, multiple myeloma and CTCL [35]. These cell lines exhibited sensitivity to ruxolitinib and vorinostat® mono-therapy, while their combination was shown to synergistically increase their inhibitory effects and induce a cessation of growth in tumor cells. This finding agrees with our results which highlight the beneficial effect of combinational ruxolitinib and resminostat therapy in inhibiting CTCL cell survival. Most importantly, resminostat

which exhibits a particular specificity towards HDAC6, in combination with ruxolitinib may present a therapeutic advantage in overcoming resistance.

Furthermore, the drugs' combinational treatment appeared most effective in inhibiting cell proliferation, as well as inducing apoptosis while increasing late apoptosis at 24h in both CTCL cell lines tested, compared to monotherapies. Our findings are in line with previous published data from Civallero *et al*. demonstrating that the combination of ruxolitinib and vorinostat® induced apoptosis in 12 cell lines of hematological malignancies, including a CTCL cell line [36]. In addition, another study showed that ruxolitinib alone enhanced apoptosis in MyLa cells, with an increase of almost 20% [20]. Our data also revealed a differential response of MyLa and SeAx cells to ruxolitinib/resminostat combinational treatment, in terms of apoptosis. Indeed, the percentage of late apoptotic MyLa cells was found to be 22% after 24h of treatment, whereas late apoptotic SeAx cells reached only 14.5%. Taken together, these data indicate a potential debulking activity for the drug combination.

Combined treatment of resminostat with ruxolitinib was shown to have a strong synergistic effect at both CTCL cell lines with a CI = 0.062 for MyLa cells and CI = 0.141 for SeAx cells. Moreover, our data revealed that the two drugs exhibit differential profiles of inhibition in terms of key signaling molecules activation in the CTCL cell lines tested, thus further confirming that MF and SS should be considered as different diseases, arising from distinct T-cell subsets [9, 34, 37]. Malignant T cells present variations in the activation of cellular signaling due to extensive crosstalk between different signal transduction pathways.

It is evident from our data that the resminostat/ruxolitinib drug combination affects the activation of Akt and JNK in both cell lines, whereas it also inhibits JAK/STAT and MAPK activation in MF cell line, a finding that further indicates the differential genetic and epigenetic mechanisms implicated in MF and SS.

In the MyLa cell line, we have shown that ruxolitinib alone or in combination with resminostat led to the inhibition of STAT3 phosphorylation, 48h post-treatment. These observations are in accordance with previous results demonstrating that STAT3 and STAT5 are constitutively expressed in CTCL [19]. We found that the combination of ruxolitinib and resminostat inhibited the phosphorylation of STAT3 and blocked JAK/STAT pathway. This could be mainly attributed to ruxolitinib's inhibitory effect on JAK2, and may also be the additive result of resminostat targeting on the JAK pathway. On the contrary, Perez *et al*. observed a marked and rapid inhibition of STAT activation with ruxolitinib that was abolished after 3h of treatment [20].

Furthermore, we showed that Akt phosphorylation was decreased after 24h of treatment with either ruxolitinib or resminostat in MyLa cell line, whereas their combination was more effective than monotherapies at 48h. The observed inhibition of Akt phosphorylation might have also contributed to the inhibition of cell proliferation and the induction of apoptosis in both MF and SS cell lines tested an issue that requires further studies, mostly due to the involvement of PI3K/Akt/mTOR pathway in autophagy and also the implication of HDAC6 in the same process. Similarly, the combination of jakitinib® and vorinostat® has been previously shown to affect autophagy in cell lines of hematological malignancies [38]. We have previously shown that activation of AKT/mTOR pathway in MF is correlated with NOTCH1, p-ERK, and p-STAT3 and is implicated in the acquisition of a more aggressive phenotype. Moreover, the combination of p-AKT, p-p70S6K, and p-4E-BP1 emerged, from our observations, as a significant potential prognostic marker in patients with advanced disease stage [35, 38]. In the CTCL cell lines tested herein, a small decrease in the phosphorylation of the downstream p-70S6K protein was observed upon treatment, indicating a possible involvement of PI3K signaling pathway, instead of mTOR.

Regarding the phosphorylation of MAP kinases in MyLa cell line, marked changes in p-ERK1/2 and p-p38 were observed only after treatment with the drugs' combination for 24h and 48h, respectively. In myeloproliferative neoplasms (MPN), p-ERK was shown to be blocked by ruxolitinib, whereas p-p38 seems to remain unaffected [37]. On the other hand, Klemke *et al*. had shown that all CTCL cell lines exhibit ERK phosphorylation 5min after TCR stimulation, but this is not detectable after prolonged (45min) TCR stimulation [39].

Furthermore, we found that the JNK phosphorylation was decreased upon drug combination at 24h and 48h of treatment, indicating the importance of JNK signaling in MyLa cells [40] and its targeting potential in MF treatment that needs to be further investigated. A previous study reported the production of high levels of VEGF through JAK3- and JNK-dependent signal pathways from malignant T cells in CTCL, thus revealing potential new therapeutic targets in CTCL [39]. On the other hand, Cardoso *et al*. have demonstrated that the incubation of MPN cells with BM-derived stroma impairs the cytotoxic action of vorinostat® and ruxolitinib combination, an effect possibly attributed to the activation of relevant survival pathways, including JNK and PI3K [40]. Therefore, inhibition of these pathways in the same system may abrogate the protective effects exerted by the stroma. These results point towards a novel therapeutic approach for the treatment of MPN patients, which relies on the dual targeting of both neoplastic clones and the microenvironmental cues.

In the SeAx cell line, the inhibition of signaling pathways upon single or combined ruxolitinib/resminostat drug treatment was less prominent. No significant inhibition of STAT3 was achieved upon drug treatment at any time point, possibly because STAT3 phosphorylation may be restored by alternative kinase families, such as MAPK and c-Jun [41]. However, ruxolitinib alone or in combination with resminostat was shown very effective in reducing p-JNK levels after 24h of treatment.

Most important, the ruxolitinib/resminostat drug combination significantly decreased p-AKT levels at 24h and 48h, in SeAx cells, indicating an important target for future treatments that needs to be further investigated. Previous studies from our group and others have also detected high levels of phosphorylated Akt in CTCL patients [35, 42–44], highlighting the importance of using AKT/mTOR signaling pathway-related targets in CTCL treatment [44, 45].

## Conclusion

Collectively, our data demonstrate for the first time the combinational inhibitory effect of ruxolitinib and resminostat in the proliferation and apoptosis of CTCL cell lines and further reveal the importance of AKT, ERK1/2 and JNK signalling as potential targets for CTCL disease management. These encouraging results are based on *in vitro* observations; therefore, they need to be further confirmed in primary patient-derived cells (cultured isolated T lymphocytes from peripheral blood and skin biopsies) in preclinical and clinical studies, validating the efficacy of this therapeutic strategy *in vitro*. Moreover, the synergistic impact of the proposed drug combination on oncogenic transformation could enable the effective use of lower doses of these agents, leading to better tolerability, and hopefully, avoidance or delay of the development of drug resistance.

## Supporting information

**S1 Fig. Testing various concentration of ruxolitinib and resminostat for cell cytotoxicity in CTCL at various concentrations at 24h.** Values are the mean ± standard deviation of three experiments. *p <0.05, statistically significant differences versus untreated and single drugs. (TIF)

## Author Contributions

**Conceptualization:** Christina Piperi, Evangelia Papadavid.

**Data curation:** Maria Dalamaga.

**Formal analysis:** Fani Karagianni, Christina Piperi, Maria Dalamaga.

**Investigation:** Fani Karagianni.

**Methodology:** Fani Karagianni, Vassiliki Mpakou, Aris Spathis, Periklis G. Foukas.

**Project administration:** Christina Piperi, Evangelia Papadavid.

**Supervision:** Christina Piperi, Evangelia Papadavid.

**Writing – original draft:** Fani Karagianni.

**Writing – review & editing:** Fani Karagianni, Christina Piperi, Vassiliki Mpakou, Vasiliki Pappa, Evangelia Papadavid.

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
