## [Decision Letter · Decision Letter 0]

11 Aug 2020

PONE-D-20-21841

Ruxolitinib with resminostat exert synergistic antitumor effects in Cutaneous T-cell Lymphoma

PLOS ONE

Dear Dr. Karagianni,

Thank you for submitting your manuscript to PLOS ONE. After careful consideration, we feel that it has merit but does not fully meet PLOS ONE’s publication criteria as it currently stands. Therefore, we invite you to submit a revised version of the manuscript that addresses the points raised during the review process.

Please revise the use of MTT assay as a cell proliferation assay. As correctly stated by the reviewer this assay is a cytotoxicity assay rather than a proliferative assay. To determine the effect on cell proliferation the authors are encouraged to preform cell cycle analyses with PI and BrdU staining and analyses of cyclin expression by immunoblots WB analyses.

We look forward to receiving your revised manuscript.

Kind regards,

Salvatore Papa, PhD

Academic Editor

PLOS ONE

Journal Requirements:

2. We noticed minor instances of text overlap with the following previous publication(s), which need to be addressed:

(1) https://onlinelibrary.wiley.com/doi/full/10.1111/jdv.16593

The text that needs to be addressed involves the Introduction section (lines 48-56).   

In your revision please ensure you cite all your sources (including your own works), and quote or rephrase any duplicated text outside the methods section. Further consideration is dependent on these concerns being addressed.

Reviewers' comments:

Reviewer's Responses to Questions

**Comments to the Author**

1. Is the manuscript technically sound, and do the data support the conclusions?

Reviewer #1: Yes

Reviewer #2: Partly

2. Has the statistical analysis been performed appropriately and rigorously? 

Reviewer #1: Yes

Reviewer #2: Yes

3. Have the authors made all data underlying the findings in their manuscript fully available?

Reviewer #1: Yes

Reviewer #2: No

4. Is the manuscript presented in an intelligible fashion and written in standard English?

Reviewer #1: Yes

Reviewer #2: Yes

5. Review Comments to the Author

Reviewer #1: The author here shows an interesting finding of synergistic effect of ruxolitinib and resminostat to inhibit proliferation and induce apoptosis on Cutaneous T-cell lymphomas. Although there are certain concerns I would like the author to address:

1. Figure 1B shows cell survival of SeAx cells around 60% at 24h whereas 80% at 48h for combination treatment with 1uM resm/15 uM rux This is surprising as mostly in case for any anticancer drug, cell death should be increasing with increase in time point, or we may say cell survival should be decreasing. Also, its contradicting to the result obtained from Apoptosis study in Figure 3, where percentage of apoptotic cells (early + late) are more in 48h (almost 60%) compared to 24 h (35%). The author should explain this discrepancy.

2. Although the author have written a detailed discussion of their finding, but it would be better for the manuscript to have a

short (conclusion section to summarize the manuscript for ease of an reader) separately following the discussion and before author contribution. Not just as a part of abstract section.

3. I would prefer the author to mention in full before using abbreviations like WGS for Whole genome sequencing or NGS for Next-Generation Sequencing, if they mean so. (Introduction section, page 11 line 68).

4. The figures representations could be improved (eg. In fig 3B and fig 3D the asterisk marks does not have consistent font size within or in comparison as well).

5. Each figure caption could start with a title in bold (to explain what this data/figure is about in short) then followed by the legend. This helps the reader to quick glance through. (eg., Fig 1. Cell viability assay of resminostat and/or ruxolitinib on CTCL. Effect of resminostat and/or ruxolitinib in cell proliferation in Myla (A) and SeAx (B) cell lines at 24h and 48h. Values are the mean � standard deviation of three experiments. *p <0.05, **p<0.001 statistically significant differences versus untreated and single drugs or combination.

Reviewer #2: This manuscript describes an interesting study, reporting the induction of apoptosis in 2 lymphoma cell lines following treatment with specific concentrations of resminostat and ruxolitinib alone and in combination. However, the manuscript is lacking 3 independent repeats of all data and despite a clear increase in efficacy when resminostat and ruxolitinib are employed in combination as opposed to their use as individual compounds, to describe the result as synergistic is incorrect as this has not been investigated in the study. Describing the MTT as a proliferation assay is also incorrect. However, the assessment of cell cycle using annexin v/PI staining and the status of proteins in a variety of cell signalling pathways, using the MILLIPLEX MAP Multi-Pathway Magnetic Bead is sound.

The authors refer to a poor survival rate (line 57) for those with lymphoma (MF). Please can the authors include some survival statistics so it is clear to the reader the current outcome for these patients. Furthermore, since the study describes putative new therapies for these patients, please can the authors include details of the current treatment strategies, why these are inadequate and therefore why new treatments are necessary. The authors report previous work revealing the importance of a variety of signalling pathways, including the JAK-STAT pathway in MF/SS, and importantly suggest the use of inhibitors targeting this pathway as a possible therapeutic strategy. However, it is unclear as to why the authors have chosen to examine the combined effect of two compounds targeting the same pathway and proteins. It would be useful to have this explained more clearly in the introduction. Since it is unlikely these compounds would be employed as a monotherapy, it would be important to establish if any synergy exists with current therapies. This should be addressed in the discussion.

Although the authors described decreased “proliferation” of the cells in response to combination treatment, in this study the MTT assay has been employed, which does not measure the proliferation rate of cells. The MTT assay is a colorimetric assay which merely reflects the number of cells present based on the activity of the mitochondria. It is a major error to refer to this data as measuring proliferation.

Although in line with previous data from the literature, the authors describe data from 1 cell line from a MF and a SS lymphoma. They report the data as an “in vitro model of CTCL”, which is misleading. It should be acknowledged that these findings need to be validated in a wider panel of cells, preferably primary patient derived cells, to confirm the efficacy of this therapeutic strategy in vitro. i.e. expand comment on line 338.

It is unclear as to the vehicle solution in which the drugs have been dissolved. Furthermore, it is not clear as to whether vehicle control treated cells have been used as the “control” cells (line 175) or are these really completely “untreated” cells (line 174). If a vehicle treated cell control has not been employed, this seriously limits the impact of the findings described throughout the study as it is wholly incorrect.

The statistics in this manuscript have been performed correctly, however, data from multiple independent repeats must be presented with standard error of the mean as opposed to standard deviation (line 163). Furthermore, some conclusions have been formed from only 2 independent repeats of experiments (line 162 “at least twice”). It is most appropriate to run 3 independent repeats of all experiments.

The authors describe a decrease in the number of cells and increased apoptosis following treatment with both resminostat and ruxolitinib, showing this drug combination represents an interesting strategy for investigation. However, in places throughout the manuscript, including the title, the authors refer to synergy between the two compounds (line 245, 339). Synergy between the inhibitors has not been shown or investigated in this manuscript. To examine synergistic or additive effects between 2 compounds, a range of concentrations of both drugs must be investigated alone and in combination using the CalcuSyn analyser of combined drug effects. It is wrong to suggest synergy has been observed in this study.

The authors refer to testing the efficacy of resminostat and ruxolitinib across a range of concentrations. Please can this data be included to explain why the concentrations of 1um and 15um, respectively, were chosen for future experiments. What was the reason for this choice of concentration?

The manuscript is written in standard English. There are some small errors for correction:

Line 197: “same wise” – please find an alternative.

Line 307: typo “\\”

Line 265: extra space

Line 267: extra comma

Throughout: et al. must be in italics

6. PLOS authors have the option to publish the peer review history of their article (what does this mean?). If published, this will include your full peer review and any attached files.

Reviewer #1: **Yes: **Dr. Arindam Pramanik

Reviewer #2: No

---

## [Author Response · Author response to Decision Letter 0]

6 Feb 2021

Dear Editor-in-Chief,

We would like to thank you and the reviewers for your thoughtful evaluation of our manuscript and for your most welcome comments/suggestions. Accordingly, we have now revised our manuscript to reflect these comments.

In the revised Text all changes/additions/modifications made in response to the Reviewers’ points are marked with track changes.

Please find below a point-by-point response to the issues raised by the Reviewers:

Academic Editor: 

Please revise the use of MTT assay as a cell proliferation assay. As correctly stated by the reviewer this assay is a cytotoxicity assay rather than a proliferative assay. To determine the effect on cell proliferation the authors are encouraged to perform cell cycle analyses with PI and BrdU staining and analyses of cyclin expression by immunoblots WB analyses.

RESPONSE: We would like to thank the academic editor for this comment. We have revised the use of MTT as a proliferation assay to cytotoxicity assay and we have also performed cell cycle analyses with 7AAD and BrdU staining. Data are shown in Figure 1. 

Due to time restriction, we have not performed cyclin expression, but we believe that 7AAD/BrdU analysis clearly demonstrated the effect of the two agents in cell proliferation.

Reviewer #1:

The author here shows an interesting finding of synergistic effect of ruxolitinib and resminostat to inhibit proliferation and induce apoptosis on Cutaneous T-cell lymphomas. Although there are certain concerns, I would like the author to address:

1. Figure 1B shows cell survival of SeAx cells around 60% at 24h whereas 80% at 48h for combination treatment with 1uM resm/15 uM rux. This is surprising as mostly in case for any anticancer drug, cell death should be increasing with increase in time point, or we may say cell survival should be decreasing. Also, its contradicting to the result obtained from Apoptosis study in Figure 3, where percentage of apoptotic cells (early + late) are more in 48h (almost 60%) compared to 24 h (35%). The author should explain this discrepancy.

Response: We have re-evaluated the effect of ruxolitinib and resminostat in cell proliferation using the 7AAD/BrdU staining and provide the new data in the revised manuscript. Based on these experiments, cell viability, cell proliferation and apoptotic data are in agreement (Figures 1 and 2).

2. Although the author have written a detailed discussion of their finding, but it would be better for the manuscript to have a short (conclusion section to summarize the manuscript for ease of an reader) separately following the discussion and before author contribution. Not just as a part of abstract section.

Response: We have included a conclusion section following the discussion.

3. I would prefer the author to mention in full before using abbreviations like WGS for Whole genome sequencing or NGS for Next-Generation Sequencing if they mean so. (Introduction section, page 11 line 68).

Response: The full descriptions of abbreviations are provided (line 76-77).

4. The figures representations could be improved (eg. In fig 3B and fig 3D the asterisk marks does not have consistent font size within or in comparison as well).

Response: The figure representations were improved and the asterisks in Figure 3 (B and D) were corrected with consistent font size.

5. Each figure caption could start with a title in bold (to explain what this data/figure is about in short) then followed by the legend. This helps the reader to quick glance through. (eg., Fig 1. Cell viability assay of resminostat and/or ruxolitinib on CTCL. Effect of resminostat and/or ruxolitinib in cell proliferation in Myla (A) and SeAx (B) cell lines at 24h and 48h. Values are the mean � standard deviation of three experiments. *p <0.05, **p<0.001 statistically significant differences versus untreated and single drugs or combination.

Response: All figure captures have been corrected as proposed by the reviewer (figure legend file).

Reviewer #2: 

This manuscript describes an interesting study, reporting the induction of apoptosis in 2 lymphoma cell lines following treatment with specific concentrations of resminostat and ruxolitinib alone and in combination. 

1. However, the manuscript is lacking 3 independent repeats of all data and despite a clear increase in efficacy when resminostat and ruxolitinib are employed in combination as opposed to their use as individual compounds, to describe the result as synergistic is incorrect as this has not been investigated in the study. 

Response: All experiments performed in triplicates and three independent repeats were performed. The authors corrected this statement at line 198-199.

We have also determined the synergistic activity of resminostat and ruxolitinib by calculating the combination index (CI). The synergistic activity of the two drugs (CI<1 for all combinations) is described in the revised manuscript [see sections Materials and Methods (line 174-183), Results (line 270-272), Discussion (line 353-355) and Figure 3. 

Dr Maria Dalamaga has performed the synergistic experiments and she has been added as a co-author to this manuscript.

2. Describing the MTT as a proliferation assay is also incorrect. However, the assessment of cell cycle using annexin v/PI staining and the status of proteins in a variety of cell signalling pathways, using the MILLIPLEX MAP Multi-Pathway Magnetic Bead is sound.

Response: ΜΤΤ is a colorimetric assay for assessing cell metabolic activity. Tetrazolium dye assays can also be used to measure cytotoxicity (loss of viable cells) or cytostatic activity (shift from proliferation to quiescence) of potential medicinal agents and toxic materials. However, the authors agreed to assess the proliferation (rapidly dividing cells) by performing cell cycle analysis with 7AAD and BrdU. The corresponding method is described in Material and Methods section (line 153-160), the data are described in Results Section (line 237-249) and in Discussion section (line 340-341). These experiments were performed by Aris Spathis and Periklis G. Foukas, and have been added as co-authors in the revised manuscript.

3. The authors refer to a poor survival rate (line 57) for those with lymphoma (MF). Please can the authors include some survival statistics, so it is clear to the reader the current outcome for these patients. 

Response: On line 64-65, statistics of the poor survival for the advanced stage disease have been added.

4. Furthermore, since the study describes putative new therapies for these patients, please can the authors include details of the current treatment strategies, why these are inadequate and therefore why new treatments are necessary. The authors report previous work revealing the importance of a variety of signalling pathways, including the JAK-STAT pathway in MF/SS, and importantly suggest the use of inhibitors targeting this pathway as a possible therapeutic strategy. However, it is unclear as to why the authors have chosen to examine the combined effect of two compounds targeting the same pathway and proteins. It would be useful to have this explained more clearly in the introduction. Since it is unlikely these compounds would be employed as a monotherapy, it would be important to establish if any synergy exists with current therapies. This should be addressed in the discussion.

Response: The authors have added a more descriptive paragraph at the beginning of the Discussion (line 303-315) and at the end of Introduction (line 125-126). 

5. Although the authors described decreased “proliferation” of the cells in response to combination treatment, in this study the MTT assay has been employed, which does not measure the proliferation rate of cells. The MTT assay is a colorimetric assay which merely reflects the number of cells present based on the activity of the mitochondria. It is a major error to refer to this data as measuring proliferation.

Response: As it has been mentioned above, the authors agreed and assessed the proliferation assay by performing cell cycle analysis with 7AAD and BrdU.

6. Although in line with previous data from the literature, the authors describe data from 1 cell line from a MF and a SS lymphoma. They report the data as an “in vitro model of CTCL”, which is misleading. It should be acknowledged that these findings need to be validated in a wider panel of cells, preferably primary patient derived cells, to confirm the efficacy of this therapeutic strategy in vitro. i.e. expand comment on line 338.

Response: In current line 432-434 (previous line 338) the phrase ‘…. in primary patient derived cells (cultured isolated T lymphocytes from peripheral blood and skin biopsies)…. validating the efficacy of this therapeutic strategy in vitro….’ was added.

7. It is unclear as to the vehicle solution in which the drugs have been dissolved. Furthermore, it is not clear as to whether vehicle control treated cells have been used as the “control” cells (line 175) or are these really completely “untreated” cells (line 174). If a vehicle treated cell control has not been employed, this seriously limits the impact of the findings described throughout the study as it is wholly incorrect.

Response: On line 139-142, a description was added where it is stated where the drugs had been dissolved. All cells that were characterized as ‘untreated’, where cells that were treated with 0.1% DMSO (vehicle control).

8. The statistics in this manuscript have been performed correctly, however, data from multiple independent repeats must be presented with standard error of the mean as opposed to standard deviation (line 163). Furthermore, some conclusions have been formed from only 2 independent repeats of experiments (line 162 “at least twice”). It is most appropriate to run 3 independent repeats of all experiments.

Response: On line 199, the authors changed the phrase ‘at least twice’ by the phrase ‘in 3 independent repeats’.

9. The authors describe a decrease in the number of cells and increased apoptosis following treatment with both resminostat and ruxolitinib, showing this drug combination represents an interesting strategy for investigation. However, in places throughout the manuscript, including the title, the authors refer to synergy between the two compounds (line 245, 339). Synergy between the inhibitors has not been shown or investigated in this manuscript. To examine synergistic or additive effects between 2 compounds, a range of concentrations of both drugs must be investigated alone and in combination using the CalcuSyn analyser of combined drug effects. It is wrong to suggest synergy has been observed in this study.

Response: The authors added the following sentence: ‘The synergistic effect between resminostat and ruxolitinib was determined by the combination index (CI) based on the Chou-Talalay method [31], as previously described [32]. The CI value was determined by the following equation: CI=sum of apoptosis of single agent treatment/apoptosis upon combined treatment. The combination effect was defined as “synergistic”, “additive” or “antagonistic” when CI was <1, 1 and >1, respectively’. A combination index (CI) of > 1 indicates antagonism, a CI of 1 denotes additivity, and a CI of < 1 indicates synergism. More specifically, CI values ranging from 0.1–0.3 are considered to indicate strong synergism, 0.3–0.7 synergism, and 0.7–0.85 moderate synergism. (line 174-183).

10. The authors refer to testing the efficacy of resminostat and ruxolitinib across a range of concentrations. Please can this data be included to explain why the concentrations of 1um and 15um, respectively, were chosen for future experiments. What was the reason for this choice of concentration?

Response: The authors have added the MTT assay with the various concentrations of both drugs to choose the concentrations for the combined experiments. Therefore, supplementary figure 1 has been added.

11. The manuscript is written in standard English. There are some small errors for correction:

Line 197: “same wise” – please find an alternative.

Line 307: typo “\\”

Line 265: extra space

Line 267: extra comma

Throughout: et al. must be in italics

Response: We have performed a grammar and spell check throughout the manuscript.

In more detail:

Previous line 197: line 262 changed to ‘similarly’

Previous line 307: line 399, the typo ‘/’ was deleted

Previous line 265: extra space on line 256 was deleted

Previous line 267: extra comma on line 359 was deleted

Et al. was changed to italics throughout the manuscript.

Trusting that we have adequately addressed the Reviewers’ concerns, we would like to thank you for your help in significantly improving our work.

With kind regards,

Fani Karagianni

Corresponding author

---

## [Decision Letter · Decision Letter 1]

24 Feb 2021

Ruxolitinib with resminostat exert synergistic antitumor effects in Cutaneous T-cell Lymphoma

PONE-D-20-21841R1

Dear Dr. Karagianni,

We’re pleased to inform you that your manuscript has been judged scientifically suitable for publication and will be formally accepted for publication once it meets all outstanding technical requirements.

Kind regards,

Salvatore Papa, PhD

Academic Editor

PLOS ONE

Additional Editor Comments (optional):

Reviewers' comments:

Reviewer's Responses to Questions

**Comments to the Author**

1. If the authors have adequately addressed your comments raised in a previous round of review and you feel that this manuscript is now acceptable for publication, you may indicate that here to bypass the “Comments to the Author” section, enter your conflict of interest statement in the “Confidential to Editor” section, and submit your "Accept" recommendation.

Reviewer #1: All comments have been addressed

Reviewer #2: All comments have been addressed

2. Is the manuscript technically sound, and do the data support the conclusions?

Reviewer #1: Yes

Reviewer #2: Yes

3. Has the statistical analysis been performed appropriately and rigorously? 

Reviewer #1: Yes

Reviewer #2: Yes

4. Have the authors made all data underlying the findings in their manuscript fully available?

Reviewer #1: (No Response)

Reviewer #2: Yes

5. Is the manuscript presented in an intelligible fashion and written in standard English?

Reviewer #1: Yes

Reviewer #2: Yes

6. Review Comments to the Author

Reviewer #1: (No Response)

Reviewer #2: (No Response)

7. PLOS authors have the option to publish the peer review history of their article (what does this mean?). If published, this will include your full peer review and any attached files.

Reviewer #1: No

Reviewer #2: No

---

## [Editor Report · Acceptance letter]

2 Mar 2021

PONE-D-20-21841R1 

­­­­­Ruxolitinib with resminostat exert synergistic antitumor effects in Cutaneous T-cell Lymphoma 

Dear Dr. Karagianni:

I'm pleased to inform you that your manuscript has been deemed suitable for publication in PLOS ONE. Congratulations! Your manuscript is now with our production department. 

Kind regards, 

on behalf of

Dr Salvatore Papa 

Academic Editor

PLOS ONE